# Intervention Effect of Non-Invasive Brain Stimulation on Cognitive Functions among People with Traumatic Brain Injury: A Systematic Review and Meta-Analysis

**DOI:** 10.3390/brainsci11070840

**Published:** 2021-06-24

**Authors:** Daniel Kwasi Ahorsu, Emma Sethina Adjaottor, Bess Yin Hung Lam

**Affiliations:** 1Department of Rehabilitation Sciences, The Hong Kong Polytechnic University, Hong Kong, China; daniel.ahorsu@connect.polyu.hk; 2Department of Behavioural Sciences, Kwame Nkrumah University of Science and Technology, Ashanti, Ghana; emmaabenaadjaottor@gmail.com; 3Department of Psychiatry, The University of Hong Kong, Hong Kong, China

**Keywords:** traumatic brain injury, cognitive functions, transcranial direct current stimulation, transcranial magnetic stimulation, non-invasive brain stimulation

## Abstract

This systematic review and meta-analysis aggregated and examined the treatment effect of non-invasive brain stimulation (NIBS) (transcranial direct current stimulation and transcranial magnetic stimulation) on cognitive functions in people with traumatic brain injury (TBI). A systematic search was conducted using databases (PubMed, Web of Science, Scopus, PsycINFO, EMBASE) for studies with keywords related to non-randomized and randomized control trials of NIBS among people with TBI. Nine out of 1790 NIBS studies with 197 TBI participants (103 active vs. 94 sham) that met the inclusion and exclusion criteria of the present study were finally selected for meta-analysis using Comprehensive Meta-Analysis software (version 3). Results showed that the overall effect of NIBS on cognition in people with TBI was moderately significant (g = 0.304, 95% CI = 0.055 to 0.553) with very low heterogeneity across studies (*I*^2^ = 0.000, Tau = 0.000). Specifically, significant and marginally significant moderate effect sizes were found for cognitive sub-domains including attention, memory, and executive function. The present findings suggest that NIBS is moderately effective in improving cognitive functions among people with TBI. In particular, NIBS may be used as an alternative and/or an adjunct treatment to the traditional approach in rehabilitating cognitive functions in people with TBI.

## 1. Introduction

Traumatic brain injury (TBI) occurs when there is a sudden trauma caused by an external force to the brain. TBI leads to clinical and functional impairment in cognition, behavioural, and sensorimotor abilities that further deteriorate one’s quality of life and life-satisfaction [1,2,3,4]. TBI can be classified based on the mechanism (open—broken fractured, or penetrated skull—or closed—blunt) [5,6] severity (mild (brief loss of consciousness for a few seconds or minutes), moderate (loss of consciousness for hours), or severe (loss of consciousness or coma for more than a day)) [6,7] and other features (e.g., location of injury) [5]. TBI is one of the growing public global health problems with more than 69 million people being affected worldwide annually. Its overall incidence was greatest in North America (1299 cases per 100,000 people) with road traffic collision and falls being the most significant contributing factors [5,8]. Given the detrimental effects of TBI, it is crucial to develop cost-effective interventions to tackle this “silent epidemic”. 

Cognitive impairments reported among people with TBI include planning, attention, memory and working memory, problem solving and decision-making, and production of language [9,10,11]. Moreover, injury to the brain might not only affect one’s cognition and motor abilities but also personality and social behaviour [9,12,13] which are also closely related to cognitive functions. Hence, there is an emergent need to develop appropriate assessment and rehabilitation programme to mitigate these challenges, particularly cognitive impairments, as early as possible to minimise the inimical effect on the TBI patients.

Technological advancements have led to improvement in assessment, diagnosis and intervention among people with TBI. Specifically, the use of magnetic resonance imaging (MRI), computed tomography (CT) scans and other latest technologies for non-invasive signal acquisition have significantly improved the quality of objective assessment (or screening) and treatment outcomes of people with TBI [14,15,16,17,18,19]. Furthermore, there are various technological applications that aid treating TBI patients. Broadly speaking, interventional strategies for TBI can be categorised into pharmacological (e.g., psychostimulants and antidepressants) [20,21] and nonpharmacological (e.g., physical therapy and cognitive behaviour therapy) [20,22] although both may be used complementarily depending on the TBI mechanism and severity. For instance, the nonpharmacological interventions such as decompressive craniectomy, early nutrition treatment, and osmotic therapy have been used to treat TBI [23]. At the post-acute phase, cognitive and physical rehabilitation (nonpharmacological) and pharmacotherapy (pharmacological) are often used in people with TBI [23]. Among all the non-pharmacological approaches, this systematic review and meta-analytic study focused on non-invasive brain stimulation (NIBS) which has been reported to have significant rehabilitation effects on people with TBI in recent years [24]. Specifically, the present study focused on two types of NIBS, namely transcranial direct current stimulation (tDCS) and transcranial magnetic stimulation (TMS). By definition, the tDCS uses low amplitude direct current (anode and cathode) to alter neuronal firing while TMS, a brief high-intensity magnetic field, uses a brief electric current through a magnetic coil to modulate neuronal activity of the brain [25,26]. The uniqueness and advantages of NIBS lie within its mechanism of intervention. That is, NIBS modulates neuronal activity without physical incision (e.g., decompressive craniectomy) which is costly, needs highly trained and experienced personnel, and leaves wound to be managed [23]. Importantly, NIBS can also be safely administered amid situations in which outpatient clinic or rehabilitation centre visits are not feasible (e.g., COVID-19). For instance, there are novel applications such as remote and/or telemedicine-based NIBS, and at-home applications of tDCS developed in view of COVID-19 [26]. As such, tDCS is not location-bounded and people with TBI or mental illness can still be treated at home with NIBS technique. Although both of NIBS techniques have been reported to have significant rehabilitation effects on people with TBI, there is no known meta-analysis that has quantified their efficacy of treating cognition specifically for TBI across studies [24,25]. However, it should be noteworthy that there was a systematic review examining the effect of NIBS on the cognitive functions of patients with stroke and TBI [27,28]. However, there is no meta-analysis to fully evaluate (accumulated effect of all findings) the effectiveness of NIBS treatment on cognitive function due to fewer studies available. If shown to be effectively alleviating the TBI symptoms, NIBS which is relatively easy to be administered and less costly can be widely adopted to treat TBI in adjunct to the traditional therapeutic approaches (e.g., computer programs, and rehearsal techniques) [29]. 

Regarding tDCS, it has been revealed to be beneficial to people with TBI with motor impairments [30,31,32], impulsivity [33], attention [34], behavioural and spatial memory impairment [35], and other clinical outcomes (e.g., mainly coma recovery and cognitive outcomes) [32,36]. Furthermore, different parts of the brain regions including prefrontal cortex and motor cortex are stimulated to treat TBI impairments such as cognitive and motor problems. For instance, among people with TBI, the prefrontal area and the dorsolateral prefrontal cortex (either left- or right-side DLPFC) was stimulated to treat non-motor (e.g., depression, attention, memory) impairment [32,34] while motor impairment was usually treated by stimulating the motor cortex [30,37,38]. Similarly, TMS has been reported to improve motor and behavioural functions after stimulating the cerebellum, motor cortex/M1, frontal (DLPFC) and/or temporal regions in TBI [39,40,41]; cognitive functions (e.g., working memory and executive function) after stimulating the DLPFC, frontotemporal, or posterior parietal cortex [41,42,43,44]; and psychological problems (e.g., posttraumatic stress disorder) after stimulating the DLPFC [40,43,45,46]. 

Prior primary studies reported divergent treatment effects (significant and non-significant) on a number of cognitive functions among people with TBI. This may have been due to the number of participants used for each primary study among others. Previous systematic review was unable to perform meta-analysis due to fewer studies [28]. Hence, the rationale of this systematic review is to use meta-analysis to accumulate the effect of all these findings (primary studies and taking that as an advantage in resolving the limited sample size issue) so as to statistically present the effect of NIBS treatment among people with TBI. In addition, this systematic review and meta-analysis examined the effectiveness of NIBS treatment on overall cognition which is broader than attention, memory, and executive function that were examined in previous study. Therefore, to help rehabilitation professionals and clinicians gain a better understanding of the NIBS interventional effect on the cognitive functions in TBI, this systematic review and meta-analysis aimed to evaluate the therapeutic efficacy of NIBS (tDCS and TMS) on cognitive functions among people with TBI. Specifically, this systematic review and meta-analysis:i.Examined the intervention efficacy of NIBS (tDCS and TMS) on cognitive functions (overall and sub-domains including attention, executive function, and memory) in TBI; andii.Examined whether potential moderators (e.g., number of session and duration) affected the efficacy of NIBS in treating different cognitive outcomes among people with TBI.

It was hypothesised that: i.Active NIBS would help improve the cognitive functions and sub-domains when compared with sham NIBS among people with TBI, andii.The moderators (number of session and duration) would affect the efficacy of NIBS in treating cognitive impairments among people with TBI.

## 2. Materials and Methods

This systematic review and meta-analysis was conducted in accordance with the Preferred Reporting Items for Systematic review and Meta-Analysis Protocols (PRISMA-P) which facilitated describing the rationale, hypothesis, and planned methods of this study although it was not prospectively registered (see Appendix A for the PRISMA checklist) [47,48]. In addition, participant, intervention, comparison and outcomes (PICO) [48,49] was adopted for the selection criteria in this study. Specifically, here are the inclusion and exclusion criteria (PICO eligibility criteria):i.Participant (P)

Studies that used participants with TBI were included in this study. These TBI participants were included regardless of their severity of treatment phase, sex, age, comorbidity and other similar demographic characteristics.

ii.Intervention (I)

The present study included studies that used NIBS specifically either TMS or tDCS techniques for interventional purposes. 

iii.Comparison (C)

Studies included in the present study should have a control group or a control task (sham) for TBI participants.

iv.Outcomes (O)

Studies that used standardised measurement tools for any of the cognitive functions as an outcome were included in the present study.

### 2.1. Search Strategy and Study Selection

Five electronic databases (PubMed, Web of Science, Scopus, PsycINFO, EMBASE) were searched using the following keywords:(“traumatic brain injur*” OR TBI OR “head injur*” OR “brain injur*” OR “brain trauma” OR concussion OR concussive) AND (tDCS OR “transcranial direct current stimulation” OR “non-invasive stimulation” OR “transcranial magnetic stimulation” OR TMS OR rTMS OR “brain stimulation”) AND (“cognitive function” OR “cognition” OR “cognitive” OR “neuropsychological” OR “neuropsychology” OR attention OR orientation OR learn* OR memory OR concentration OR “mental-process*” OR “executive function*” OR visuospatial OR language OR intelligence OR “intellectual function*” OR “motor function” OR cogniti* OR “visual-spatial” OR “visuo-spatial” OR recall OR recognition OR “problem solving” OR “reaction time” OR vigilance OR reason* OR psychomotor OR motor OR processing OR planning OR “verbal fluency” OR inhibit*).

English language restriction was applied, and minor modification of the search terms were made with respect to each database. All studies published from inception of each of five databases until 10 October 2020. The titles and abstracts of identified articles were first checked for suitability and included if it meets the PICO criteria above. The full text of all potential studies was then retrieved to assess them for the full PICO eligibility criteria. The literature search and study selection was performed by D.A. and E.A. with the supervision of B.L. 

### 2.2. Assessment for Quality of Reporting, Methodological Quality and Risk of Bias

In order to ensure that our findings were valid and they were from studies that had robust methodological and analytical qualities, the Joanna Briggs Institute’s (JBI) critical appraisal checklist for quasi-experimental studies (non-randomized experimental studies), and randomized controlled trials were used. These tools assessed the quality and biases of the studies [50,51]. For each study that was examined, the number of items responded “yes” by the reviewers (D. A and E. A) out of the total number of items were counted. However, “not applicable” criteria were excluded, while those items that were responded as “unclear” were treated as “no”, which implied that they did not meet the quality criteria. Hence, a study was awarded an overall quality rating of “low” if it had a “yes” score of 0–33% of the JBI questions, “medium” if it had a “yes” score of 34–66% of the JBI questions, and “high” if it had a “yes” score of 67% or more of the JBI questions (see Table 1). Studies that were of low quality were excluded from the present meta-analysis using the JBI criteria which was adopted in previous systematic review [52]. Regarding the risk of bias assessment, the Cochrane’s risk of bias tool was used [53] and the bias assessment was done across seven domains which included random sequence generation, allocation concealment, blinding of participants (checking for possible performance bias), blinding of outcome assessment (checking for possible detection bias), incomplete outcome data, selective reporting, and other biases. These seven item domains were assessed using three risk of bias categories: low, high and unclear bias. The overall bias of a study was reported as high if majority of seven item domains were responded as high or unclear. Otherwise, they were reported as low (See Table 1) [54]. 

### 2.3. Data Extraction

Outcome measures of the selected studies such as trail making test (A and B; TMT-A and B), Rey auditory verbal learning test (RAVLT), and brief visuospatial memory test (BVMT) were coded into different cognitive sub-domains using the Compendium of Neuropsychological Tests [54], accepted neuropsychological categorization as per previous studies [62], or by consensus between D.A. and E.A. Salient information such as the means and standard deviations (pre- and post-test) were extracted from the RCT and quasi studies for both active and sham groups using a set test–retest correlation of 0.6. Few studies were entered as post-training mean change or Hedges’ g. All of these extracted data were entered into Comprehensive Meta-Analysis version 3 (CMA, Biostat, Englewood, NJ, USA) for the meta-analysis.

### 2.4. Data Analysis 

The main outcome was standardized mean difference (SMD, calculated as Hedges’ *g*) of change from baseline to post-intervention between active and sham groups. Analyses were conducted for overall cognition and for each of the following cognitive domains: memory, attention, and executive functions. Analyses of global cognition, motor function, social perception and language were not performed due to insufficient numbers of studies reporting these outcomes—1, 1, 1, and 2 respectively. Precision of the SMD was estimated using 95% confidence intervals (CI). A positive SMD implies a better intervention effect among the active group compared to the sham group.

The analyses were performed in two main stages. The first stage involved combining all cognitive outcomes from each study and pooling them to determine their overall efficacy in enhancing cognition in TBI. The second stage involved domain-specific meta-analyses, in which only studies that reported specific cognitive outcome domains were included, using one combined SMD per study. The analyses (pooling outcomes) were corrected for inter-correlation across outcomes using a correlation of 0.7 [60,62,63,64] and the random-effects model. Specifically, a random-effects model was used because it attempts to generalise findings beyond the included studies by assuming that the selected studies are random samples from a larger population [63,64]. Hence, Hedges’ *g* SMDs of ≤0.30, >0.30 and <0.60, and ≥0.60 were considered as small, moderate, and large, respectively based on the same convention for description of Cohen’s *d* effect sizes [60,62]. Furthermore, the heterogeneity across studies was assessed using the *I*^2^ statistic with values of 25, 50, and 75% indicating low, moderate, and large heterogeneity, respectively [60,62]. Funnel plots were used to assess the publication bias [59,60]. Potential moderators (number of treatment session and duration) were identified and analysed using meta-regression [61,63]. Sensitivity analyses were performed taking into consideration the study’s design and risk of bias in order to ascertain the robustness of the observed outcomes [57]. All analyses were performed using CMA by D.A. and E.A. who are trained researchers.

## 3. Results

### 3.1. Study Selection

Out of 1790 articles found in the initial search, 816 were initially screened for eligibility based on published title and abstract. Seven ninety-nine articles were excluded based on the inclusion-exclusion criteria of the present study with 17 full-text articles further assessed for eligibility. After assessing their eligibility, nine studies (see Figure 1) were deemed eligible for inclusion in this systematic review and meta-analysis based on the aforementioned criteria.

### 3.2. Characteristics of Included Studies

Nine NIBS studies were generally included in this study which encompass eight randomized control trials (RCTs) and a quasi-experimental study (three anode tDCS and six repetitive (r) TMS). In addition, these NIB interventions were comprised of different numbers of sessions (3 to 20) and durations (averagely 10 to 30 min/day). In these nine studies, 197 participants (103 active vs. 94 sham) with various cognitive functions (attention, executive function, memory, global cognition, motor function, social perception and language) were included. Meta-analysis was performed on all of these nine studies in the present study. 

### 3.3. Intervention Effect on Overall Cognitive Outcomes

In general, there was a significant positive and moderate effect of active NIBS on overall cognitive outcomes (*k* = 9, *g* = 0.304, 95% CI 0.055 to 0.553, *p* = 0.017; see Figure 2). Heterogeneity across studies was very low (Q = 5.116, *df* = 8, *p* = 0.745, *I*^2^ = 0.000, Tau = 0.000). The funnel plot result did not show significant asymmetry (Figure 3).

### 3.4. Intervention Effect on Domain-Specific Outcomes

#### 3.4.1. Attention

There was a significant positive and moderate effect of active NIBS on attention (*k* = 7, *g* = 0.316, 95% CI 0.022 to 0.610, *p* = 0.035; see Figure 4). Heterogeneity across studies was very low (Q = 2.587, *df* = 6, *p* = 0.859, *I*^2^ = 0.000, Tau = 0.000). The funnel plot result did not show significant asymmetry.

#### 3.4.2. Memory

A significant positive and moderate effect was found for active NIBS on memory (*k* = 6, *g* = 0.319, 95% CI 0.005 to 0.634, *p* = 0.047; see Figure 5). Heterogeneity across studies was very low (Q = 1.542, *df* = 5, *p* = 0.908, *I*^2^ = 0.000, Tau = 0.000). The funnel plot result did not show significant asymmetry.

#### 3.4.3. Executive Function

A moderate and positive effect was found for active NIBS on executive function, but the effect was only marginally significant (*k* = 6, *g* = 0.318, 95% CI −0.017 to 0.653, *p* = 0.063; see Figure 6). Heterogeneity across studies was very low (Q = 6.698, *df* = 5, *p* = 0.594, *I*^2^ = 0.000, Tau = 0.000). The funnel plot result did not show significant asymmetry.

#### 3.4.4. Other Cognitive Domains

Studies that evaluated fluid and crystallised cognitive abilities [55] and global cognition [56] among people with TBI revealed no significant difference between the active and sham groups after rTMS intervention. In addition, there were no between-group differences on social perception [65] or language abilities [66] among people with TBI after using anode tDCS when compared with the sham. 

### 3.5. Identification and Analysis of Potential Moderators

The number of sessions ranged from three to twenty and only four studies reported the minutes taken per session (i.e., 10–30 min per day). Hence, only the number of sessions was added as the moderator in the analysis because most studies did not report each session duration. The meta-regression indicated that the number of treatment sessions was not significantly related to the effect size on overall cognition (Q = 2.14, *df* = 1, *p* = 0.143; β = −0.034, *p* = 0.143). No meta-regression was performed on the sub-domains due to the limited number of primary studies.

### 3.6. Sensitivity Analyses

Motes, et al. [66] was first removed from the initial nine studies as it used a non-randomised control trial. The meta-analysis on the eight studies revealed a significant positive and small effect of active NIBS on overall cognitive outcomes (*k* = 8, *g* = 0.292, 95% CI 0.033 to 0.550, *p* = 0.027; see Appendix A). 

Thereafter, Motes, et al. [66] and Leung, et al. [58] were both removed from the analysis as they had high risk of bias or could not be assessed (not applicable). The meta-analysis on the seven studies revealed a non-significant effect of active NIBS on overall cognitive outcomes (*k* = 7, *g* = 0.251, 95% CI −0.021 to 0.524, *p* = 0.071; see Appendix A). 

## 4. Discussion

Given the detrimental impacts of cognitive impairments associated with TBI, this study examined the interventional effect of NIBS on cognitive functions among people with TBI. The findings of this systematic review and meta-analysis showed that NIBS was moderately effective in improving the overall cognition and the sub-domains—attention, memory and executive function in TBI. Moreover, this positive interventional effect was not affected by the moderators, specifically the number of treatment sessions. These findings suggest that NIBS can be used as an alternative and/or an adjunct treatment to the traditional approach in rehabilitating cognitive functions (e.g., computer programs, and rehearsal techniques) in people with TBI which have significant clinical implications [28]. More importantly, NIBS might be adopted to effectively treat people with TBI even amid COVID-19 pandemic in which visiting outpatient clinics and rehabilitation centres is not as feasible [25]. 

The meta-analytic finding indicated that NIBS was effective in moderately improving the general cognition of people with TBI. This suggests that NIBS treatment is effective to improve the overall cognitive function of people with TBI which might in turn enhance their psychosocial well-being [9,12,13]. Future studies may empirically examine the NIBS treatment effect of TBI’s psychosocial functions. In addition, it was noted that majority of the studies with different cognitive outcomes analysed in the present study stimulated the DLPFC which signifies the multi-functionality (e.g., executive function, memory) of this brain region. In fact, previous reviews also reported its significant involvement in treating other non-cognitive problems [9,12,13,24]. Moreover, the significant aggregated interventional NIBS effect found in the current study is noteworthy as only one primary study alone [42] showed significant effect of NIBS on cognition while the other eight individual primary studies found no significant treatment effect [55,58,67,68]. This suggests that with an increase in sample size, significant NIBS interventional effect would have been found between the active and sham groups in treating cognitive functions among TBI. 

Two potential moderators (number of sessions and duration of each session) were identified among those nine primary studies and neither of them were found to be significant in the present study. As such, the number of treatment sessions do not have anything to do with the efficiency of NIBS treatment on cognition among people with TBI. With more primary studies to be analysed, future meta-analysis may examine the role and efficacy of these moderating variables as previous reviews suggested [60,69]. Other factors that may have contributed to the non-significant between-group difference in previous individual studies include varied aetiological complexities (e.g., violent blow, vehicular collision, and falls) and severities (e.g., mild, moderate, and severe) of TBI, number of treatment sessions, duration of each session, and other methodological procedures (e.g., lateral vs. bilateral TMS) [24,60,70]. Furthermore, the heterogeneity results (Q-value less than the degrees of freedom) indicated that there was no evidence that the effect size varied across TBI individuals. This finding implies that the impact of NIBS intervention might be the same among TBI population. Taking all these findings and the quality of the studies included into consideration, it is concluded that NIBS intervention is effective in improving the overall cognitive functions (particularly attention, memory and executive function) among people with TBI.

With regard to the cognitive domain-specific efficacy, the pooled analysis revealed that NIBS is effective in moderately improving attention, memory and executive functioning after stimulating the DLPFC. This finding is consistent with prior literature showing that DLPFC is a multifaceted hub for treating attention, memory, and executive functioning impairment in TBI [24,25]. Specifically, the present study reported moderate effect sizes which signifies the positive intervention efficacy of TMS and tDCS in improving the attentional and memory abilities in TBI. The effect size on executive function abilities was moderate but marginally significant, which might be because of insufficient primary studies on executive function using TMS or tDCS [24,71]. Other domain-specific analyses were not conducted for fluid and crystallised cognitive abilities, global cognition, social perception, and language abilities due to limited number of studies reporting on those sub-domain variables. Nevertheless, based on our current findings and previous studies [24,25] NIBS stimulation at the PFC would have had positive effect on these cognitive sub-domains. Therefore, there may be a need for more NIBS studies in these cognitive sub-domains which can give us a holistic view of NIBS intervention efficacy in TBI. Furthermore, the heterogeneity results (Q-value was less than the degrees of freedom for each of the sub-domains) implied that the significant impact of NIBS intervention on each of these sub-domains would be the same among TBI population. Taken all these findings together, NIBS intervention is effective in improving the overall and cognitive sub-domains (attention, memory, and executive function) across different people with TBI.

### Limitations and Future Directions

There is a number of limitations which might be addressed in future studies. First, this study included nine primary studies for the meta-analysis which limited further analysis to be performed as per the rule of thumb for meta-regression [61,63]. This was compounded by most of the studies’ failure to report some of the salient information required for meta- analysis. In addition, the limited number of studies reporting the duration of each session prevented us from analysing the effect of session duration on cognition and sub-group analysis (tDCS vs. TMS) was not performed because of similar reason. Future primary studies (e.g., RCTs) may therefore compare variables such as the number of sessions and duration of sessions to get a vivid understanding about the effect of these variables on cognition. Second, all the primary studies had limited sample sizes which might have contributed to their non-significant findings. Hence, future studies may consider increasing the sample size appropriately. Third, the primary studies had a number of confounding variables which might have contributed to their nonsignificant findings. Therefore, future primary studies may possibly control potential confounding factors in order to obtain a more stringent treatment effect. It also implies that rehabilitation professionals should carefully take into consideration the nature and severity of cognitive challenges experienced by the people with TBI in order to design and implement appropriate NIBS interventions for them. Fourth, more NIBS studies on fluid and crystallised cognitive abilities, global cognition, social perception, and language abilities in TBI is needed so to provide a holistic view of NIBS intervention effect in this group of people. Last but not least, future NIBS studies may incorporate neuroimaging methods such as functional and structural magnetic resonance imaging (MRI) [15,18,19] as one of the outcome measures so as to provide both anatomical and functional visualizations to help explain the mechanism by which NIBS alleviates cognitive functions among people with TBI.

## 5. Conclusions

This study is the first to aggregate existing interventional findings of NIBS on cognitive functions (overall and domain-specific cognitive functions such as attention, memory and executive function) among people with TBI. More importantly, the present findings have made a noteworthy contribution to the field of clinical neuroscience. Specifically, it is revealed that NIBS (with majority of the primary studies stimulating the DLPFC) may improve the cognition abilities and its domain-specific functions such as attention, memory and executive function among the people with TBI. Moreover, the number of sessions of NIBS was not found to be a significant moderator in the present study, suggesting a positive and significant NIBS interventional effect on cognition in TBI regardless of the number of treatment sessions. Nevertheless, it is strongly suggested that further studies (e.g., RCTs) ascertain the veracity of these variables (e.g., aetiological complexities and severities of TBI, number/duration of sessions and the methodological procedures adopted in the NIBS intervention) in the NIBS intervention effect. In addition, future NIBS studies may add neuroimaging techniques as one of the outcome measures so as to provide both anatomical and functional visualizations to help explain the mechanism by which NIBS alleviates cognitive functions among people with TBI. Taken all the findings together, clinicians and professional stakeholders may administer NIBS to improve the cognitive challenges especially attention, memory abilities and executive functions encountered by people with TBI even during COVID-19 (e.g., home-based tDCS). Although we might also need to be cautious about the significant positive effect of NIBS found in the present study with a small sample of studies, NIBS (e.g., remote and/or telemedicine-based NIBS) might provide an alternative treatment for TBI individuals amid COVID-19 when visits at outpatient clinics or rehabilitation centers are not as feasible. 

## Figures and Tables

**Figure 1 brainsci-11-00840-f001:**
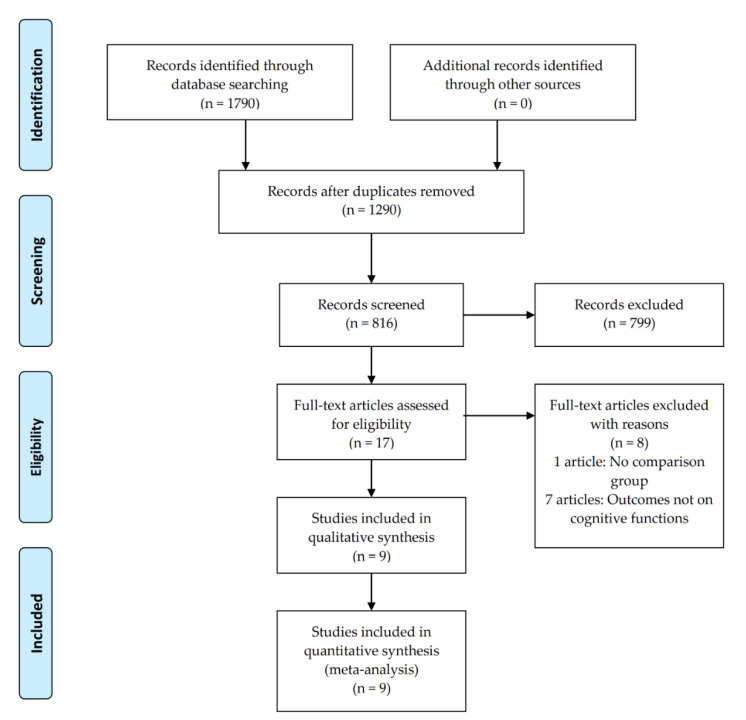
Flowchart of literature identification and selection.

**Figure 2 brainsci-11-00840-f002:**
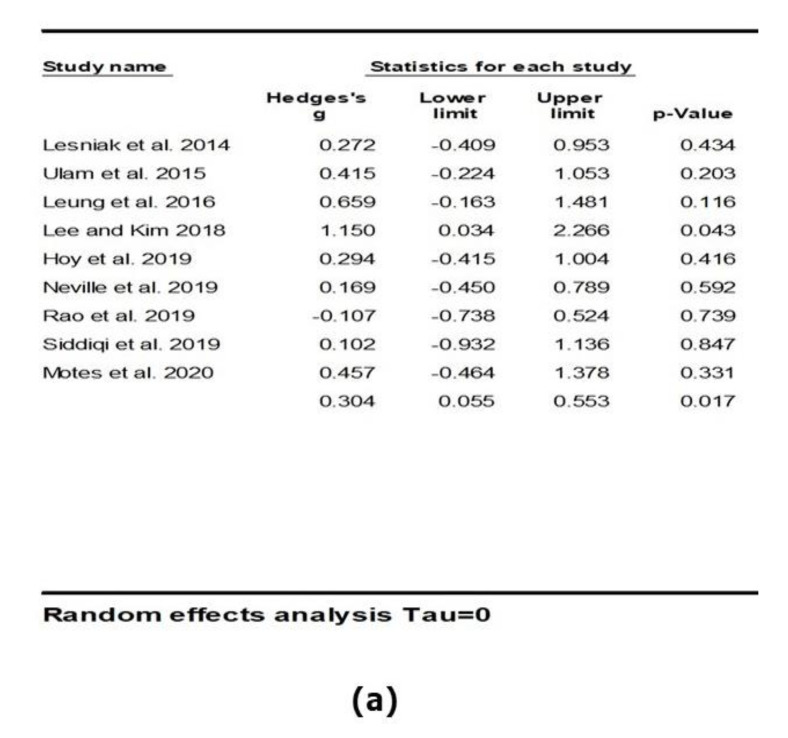
The effect of non-invasive brain stimulation (NIBS) on overall cognitive outcomes; (**a**) summary table for each study; (**b**) forest plot for each study [38,40,55,56,57,58,59,60,61].

**Figure 3 brainsci-11-00840-f003:**
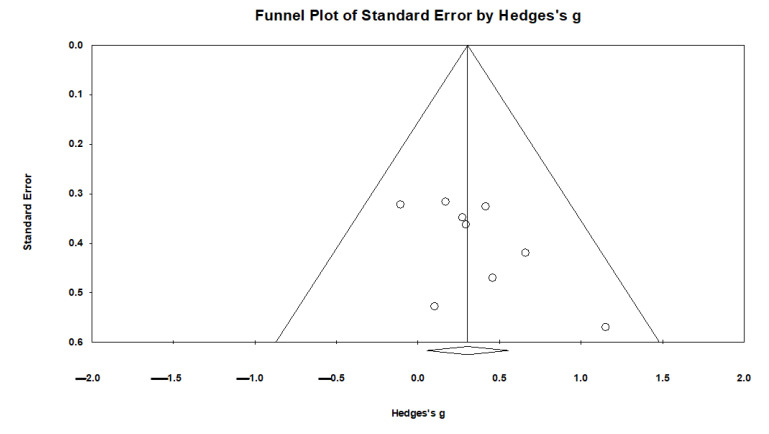
The effect of non-invasive brain stimulation (NIBS) on overall cognitive outcomes: a funnel plot.

**Figure 4 brainsci-11-00840-f004:**
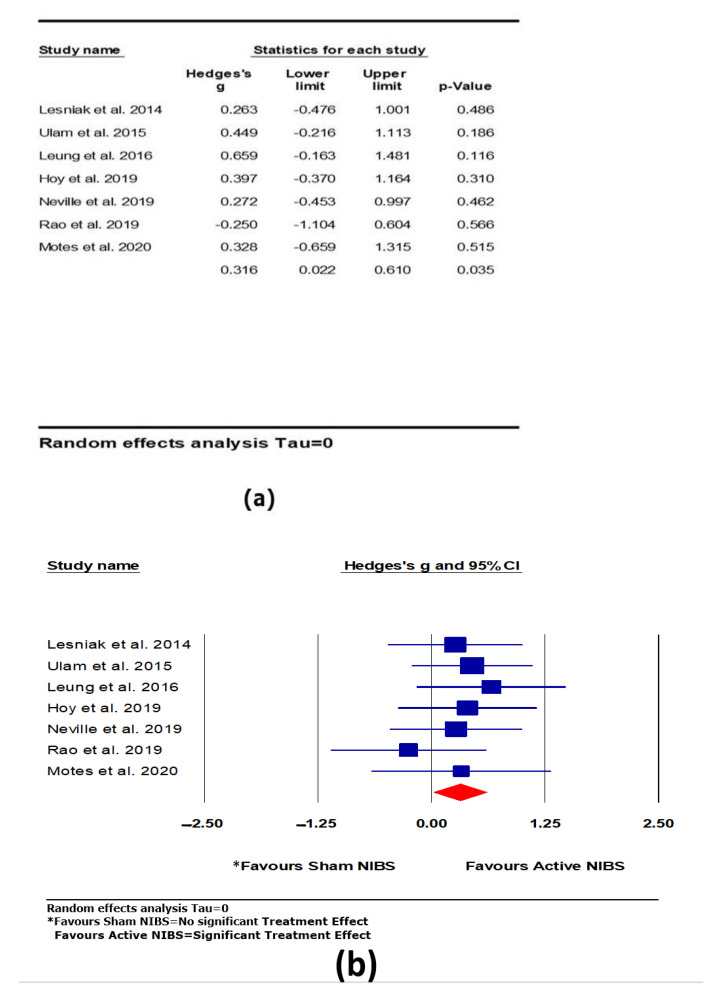
The effect of non-invasive brain stimulation (NIBS) on attention; (**a**) summary Table for each study; (**b**) forest plot for each study [38,55,56,57,58,59,61].

**Figure 5 brainsci-11-00840-f005:**
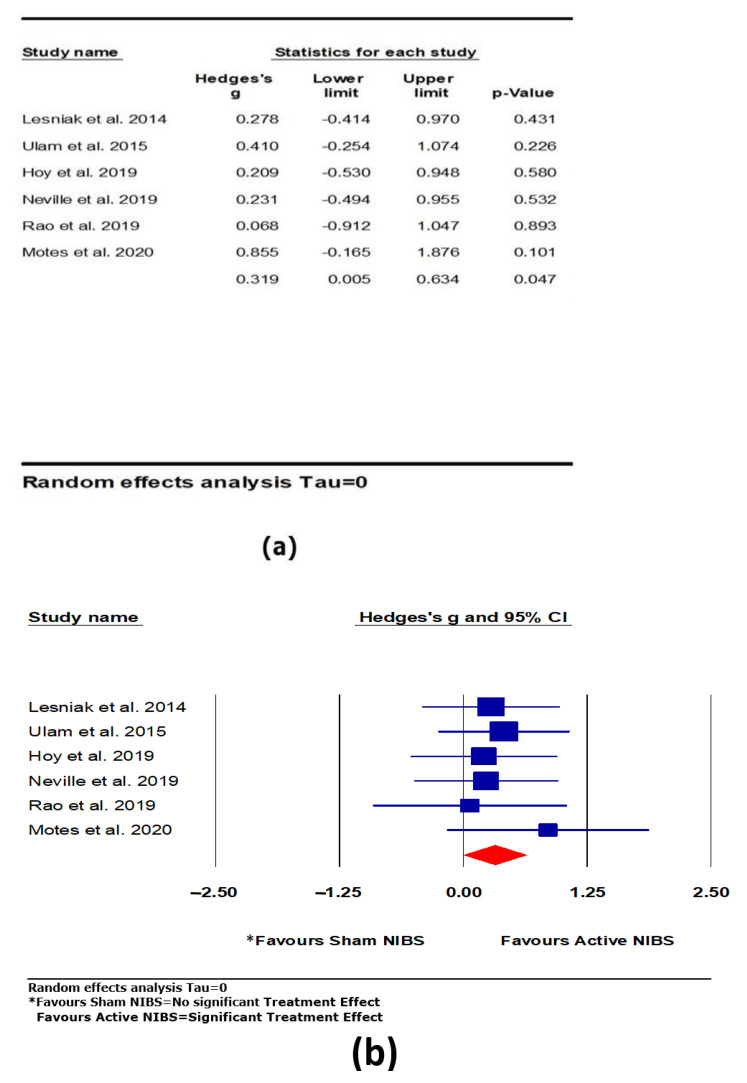
The effect of non-invasive brain stimulation (NIBS) on memory; (**a**) summary table for each study; (**b**) forest plot for each study [38,55,56,57,58,59,61].

**Figure 6 brainsci-11-00840-f006:**
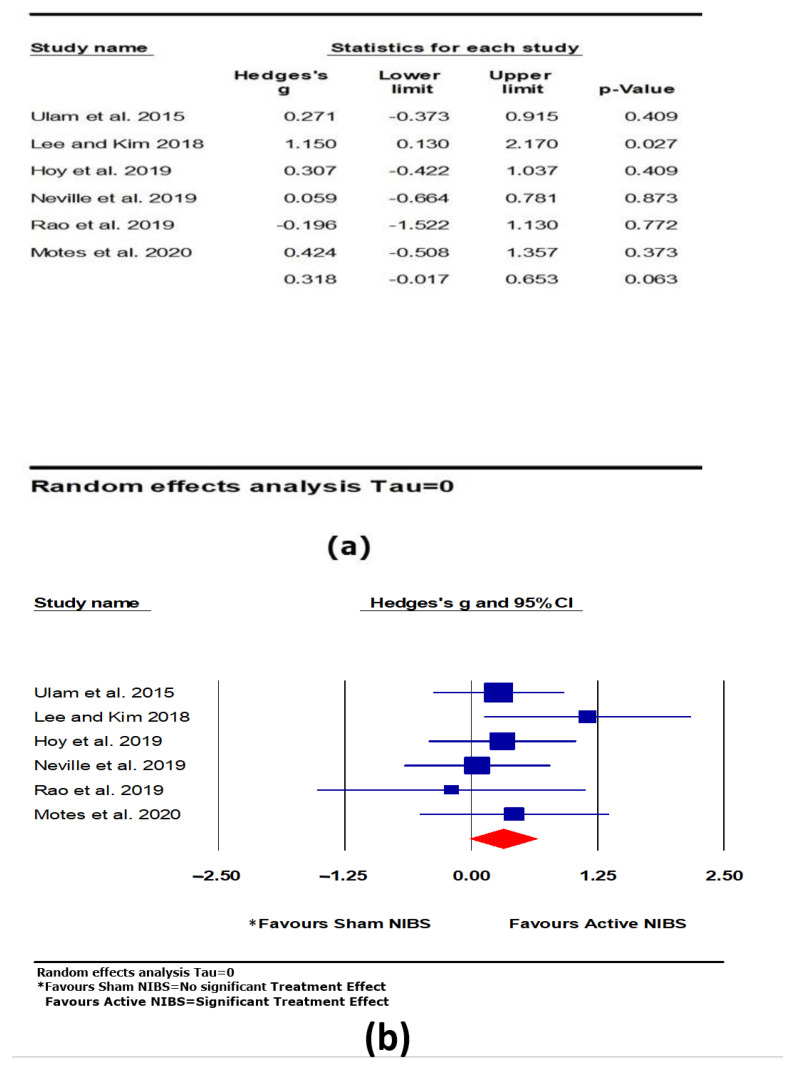
The effect of non-invasive brain stimulation (NIBS) on executive function; (**a**) summary table for each study; (**b**) forest plot for each study [38,40,57,58,59,61].

**Table 1 brainsci-11-00840-t001:** Characteristics of the included studies.

Author and Year	N (Active, Sham)	Age Active, Age Sham	Study Design	Treatment Type	No of Sessions	Duration (min/day)	Outcomes	Quality Appraisal	Risk of Bias
Hoy et al. (2019) [38]	21 (11, 10)	41.27 ± 10.04, 51.80 ± 13.38	RCT	Bilateral dlPFC rTMS	20	― 20 days	Attention Executive function Memory	High	Low
Lee and Kim (2018) [40]	13 (7, 6)	42.42 ± 11.32, 41.33 ± 11.02	RCT	Right dlPFC rTMS	10	30 min/day	Executive function	High	Low
Lesniak et al. (2014) [55]	23 (12, 11)	28.3 ± 9, 29.3 ± 7.7	RCT	Left dlPFC Anode tDCS	15	10 min/day	Attention Memory	High	Low
Leung et al. (2016) [56]	24 (12, 12)	41 ± 14, 41 ± 12	RCT	LMC rTMS	3	― 3 days ^a^	Attention	High	High
Motes et al. (2020) [57]	15 (9, 6)	40.9 ± 5.0, 40.8 ± 10.9	Non-RCT	preSMA/dACC Anode tDCS	10	20 min/day	Attention Executive function Memory Language	High	N/A
Neville et al. (2019) [58]	30 (17, 13)	32.62 ± 12.81, 29.0 ± 10.35	RCT	Left dlPFC rTMS	10	― 10 days	Attention Executive function Memory Motor function	High	Low
Rao et al. (2019) [59]	30 (13, 17)	39.8 ± 14.2, 40.2 ± 14.6	RCT	LFR dlPFC rTMS	20	― 20 days	Global cognition Attention Executive function Memory	High	Low
Siddiqi et al. (2019) [60]	15 (9, 6)	43 ± 13, 50 ± 18	RCT	Bilateral dlPFC rTMS	20	― 20 days ^b^	Fluid cognition Crystallised cognition Overall cognition	High	Low
Ulam et al. (2015) [61]	26 (13, 13)	31.34 ± 9.8, 35.70 ± 14.7	RCT	Left dlPFC Anode tDCS	10	20 min/day	Attention Executive function Memory Social perception	High	Low

dACC: dlPFC: dorsolateral prefrontal cortex; LFR: low-frequency right-sided; LMC: left motor cortex; preSMA/dACC: presupplementary motor area/dorsal anterior cingulate cortex. ^a^ Three days with at least 24 h or no more than 72 h apart. ^b^ Exact days not given.

## Data Availability

The study did not report any data.

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
