# Peer review of "Intervention Effect of Non-Invasive Brain Stimulation on Cognitive Functions among People with Traumatic Brain Injury: A Systematic Review and Meta-Analysis"

_brainsci, 2021, doi:10.3390/brainsci11070840_

Round 1

Reviewer 1 Report

This systematic review of the literature examines NIBS for patients with TBI. There are some methodological issues and I have the following major and minor comments for the authors to consider.

Specific comments:

  1. "A systematic search was conducted using databases (e.g., PubMed)" - please be more specific here.
  2. Please change "Given the detrimental impact that brought about by TBI" to "Given the detrimental effects of TBI".
  3. "... there is no known meta-analysis that has quantified their efficacy of treating cognition in TBI across studies [22,23]" - there is actually a systematic review study prospectively registered with the PROSPERO database of systematic reviews (CRD42020183298). See: pubmed.ncbi.nlm.nih.gov/33807188. The rationale for the current review must be better justified and explained in light of this.
  4. "... effectively enhancing the TBI symptoms" - is enhancing the correct word choice here??
  5. The PRISMA checklist was not included in the submission.
  6. As per PRISMA guidelines, please specify in the methods section if the review protocol was prospectively registered. Indicate if a review protocol exists, if and where it can be accessed (e.g., Web address).
  7. The reasons for exclusion (at each step) should be clearly stated in Figure 1.
  8. The authors mentioned random-effects model was adopted. The authors should explain why random-effects model was used, for example, random-effects model attempted to generalize findings beyond the included studies by assuming that the selected studies are random samples from a larger population (citation: pubmed.ncbi.nlm.nih.gov/31602169).
  9. No sensitivity analysis was performed.
  10. What are the "traditional approach in rehabilitating cognitive functions"? This was repeated twice in the manuscript with no further elaboration. More information is necessary.
  11.  It is clear that the rehabilitation literature in this area of field is still lacking depth and breadth. Some areas for future work should be suggested, in addition to the obvious fact that well designed and conducted studies with clinical power are needed to answer these clinically relevant questions.

Reviewer 2 Report

In this article Authors examined the treatment effect of Non- invasive Brain Stimulation transcranial direct current stimulation and transcranial magnetic stimulation on cognitive functions in people with Traumatic Brain Injury. A systematic search was conducted using databases for studies with keywords related to non-randomized and randomized control trials of NIBS among people with TBI. 

My comments to the article are as follows:

- As part of the Introduction, I propose to provide a broader background in the field of sources of the formation of brain potentials. I propose to refer, for example, to: Characteristics of Question of Blind Source Separation Using Moore-Penrose Pseudoinversion for Reconstruction of EEG Signal, Automation 2017: Innovations In Automation, Robotics And Measurement Techniques, Book Series: Advances in Intelligent Systems and Computing, Springer, 2017. I also propose to refer to the information that devices for testing the state of relaxation / activity of the human brain, for example visualizing these changes with diodes, have already been developed (as an example of non-invasive signal acquisition, although based on EEG, but will give a broader look at data acquisition). For example, reference may be made to: Project and Simulation of a Portable Device for Measuring Bioelectrical Signals from the Brain for States Consciousness Verification with Visualization on LEDs, Challenges In Automation, Robotics And Measurement Techniques, Book Series: Advances in Intelligent Systems and Computing, Springer.

- I propose to draw the algorithm in Fig. 1 in accordance with the methodology of writing algorithms.

- Please explain in detail how the data analysis was carried out.

- The charts in Figures: 2 to 6 are not very clear. I propose to separate these tables from the graphs. As part of the graphs, the description of the axis should be clarified.

- As part of the Conclusions, please write about plans for the future in the field of research.

- The authors of the work should also be completed in the article. Please enter your data and affiliations after the title of the article.

Additional editorial note:

- Tables are not formatted according to the MDPI format (both Table No. 1 and those included in the pictures). This needs to be corrected. They should also be put in order. The article looks unhumanly formatted.

Round 2

Reviewer 1 Report

Thank you for the revisions.

Specific comments:

  1. The rationale for the present study was only weakly justified.
  2. Please standardize the font and colour choices in Figures 2-4.
  3. Note that Figure 2 was not labeled.

Reviewer 2 Report

Dear Authors, 

Regarding your reply to my comment No. 1, I believe that it is not fully correlated with my request. Please note that I have noticed that there is no background regarding "in the field of sources of the formation of brain potentials.". In this regard, reference can be made, for example, to: "Characteristics of Question of Blind Source Separation Using Moore-Penrose Pseudoinversion for Reconstruction of EEG Signal, Automation 2017: Innovations In Automation, Robotics And Measurement Techniques, Book Series: Advances in Intelligent Systems and Computing , Springer, 2017 ". I believe that a skilful reference to this range will be valuable for this work.

Further, you did not refer to the second part of my comment regarding: "I also propose to refer to the information that devices for testing the state of relaxation / activity of the human brain, for example visualizing these changes with diodes, have already been developed ( as an example of non-invasive signal acquisition, although based on EEG, but will give a broader look at data acquisition). ".

The text fragment in the article you refer to (added to the article as part of the reply to Comment # 1) is not cited.

My next comment was similarly incomprehensible. My point is that the flowchart in Fig. 1 should be formatted according to the idea of ​​building a flowchart. This way you don't build flowcharts. Flowchart is a certain scheme of an algorithm as a procedure. I meant the algorithm as a procedure, not the algorithm used as a tool.

The tables are still connected by charts. My point was that this should be divided. Separately charts. Separate tables.
